# The Salinity Survival Strategy of *Chenopodium quinoa*: Investigating Microbial Community Shifts and Nitrogen Cycling in Saline Soils

**DOI:** 10.3390/microorganisms11122829

**Published:** 2023-11-21

**Authors:** Xuli Zhao, Tianzhu Meng, Shenghan Jin, Kaixing Ren, Zhe Cai, Bo Cai, Saibao Li

**Affiliations:** 1College of Agricultural Science and Engineering, Hohai University, No. 8 Focheng West Road, Nanjing 211100, China; 2College of Water Resources and Civil Engineering, Tibet Agricultural and Animal Husbandry University, No. 8 Xueyuan Road, Linzhi 860000, China

**Keywords:** *Chenopodium quinoa*, saline soils, microbial community, nitrogen transformations, salt tolerance

## Abstract

Quinoa is extensively cultivated for its nutritional value, and its exceptional capacity to endure elevated salt levels presents a promising resolution to the agricultural quandaries posed by salinity stress. However, limited research has been dedicated to elucidating the correlation between alterations in the salinity soil microbial community and nitrogen transformations. To scrutinize the underlying mechanisms behind quinoa’s salt tolerance, we assessed the changes in microbial community structure and the abundance of nitrogen transformation genes across three distinct salinity thresholds (1 g·kg^−1^, 3 g·kg^−1^, and 6 g·kg^−1^) at two distinct time points (35 and 70 days). The results showed the positive effect of quinoa on the soil microbial community structure, including changes in key populations and its regulatory role in soil nitrogen cycling under salt stress. *Choroflexi*, *Acidobacteriota*, and *Myxococcota* were inhibited by increased salinity, while the relative abundance of *Bacteroidota* increased. *Proteobacteria* and *Actinobacteria* showed relatively stable abundances across time and salinity levels. Quinoa possesses the ability to synthesize or modify the composition of keystone species or promote the establishment of highly complex microbial networks (modularity index > 0.4) to cope with fluctuations in external salt stress environments. Furthermore, quinoa exhibited nitrogen (N) cycling by downregulating denitrification genes (*nirS*, *nosZ*), upregulating nitrification genes (Archaeal *amoA* (AOA), Bacterial *amoA* (AOB)), and stabilizing nitrogen fixation genes (*nifH*) to absorb nitrate–nitrogen (NO_3_^−^_N). This study paves the way for future research on regulating quinoa, promoting soil microbial communities, and nitrogen transformation in saline environments.

## 1. Introduction

Soil salinization is a widespread environmental stressor that adversely affects plant growth and soil health [1]. It is estimated that salt-affected soil covers approximately 8.7% of Earth’s land area [2]. The low permeability of saline soils directly inhibits the structure and metabolic activity of the soil microbial community [3,4]. It greatly impacts microbial-mediated soil ecological functions and a range of processes related to soil fertility [5]. Additionally, salinization slows down or inhibits various nitrification processes in the soil, leading to the accumulation of nitrite or accelerated loss of ammonia–nitrogen (NH_4_^+^) through volatilization [6]. This disruption disrupts the normal nitrogen (N) transformation in the soil and affects plant uptake and the utilization of N, as well as soil microbial activity [7].

Quinoa (*Chenopodium quinoa*), renowned for its ability to grow in extreme environments and high nutritional value [8], serves as an ideal crop model for studying salt stress tolerance through interactions among microorganisms, plants, and soil [9,10,11]. Studies reveal that quinoa can enhance soil microbial diversity and functionality in saline ecosystems [11,12,13]. Therefore, quinoa can be utilized to rebuild the composition and functionality of microbial communities in saline soils, aiming to enhance and safeguard the essential physicochemical and biological properties of salt-affected soils [14].

Keystones have been demonstrated to have the potential to promote nutrient transformation and plant growth [15]. Furthermore, the biodiversity of keystones largely determines the functional capability of soil microbial communities [16]. Hence, focusing on keystones offers the potential to organize information regarding microbial interactions more effectively, simplify microbiome analysis, and identify the critical components of individual genera within microbial communities [17,18]. Microbial communities function as keystones that regulate the interactions and functional aspects of microbial community dynamics [19,20]. For instance, the nutrient cycling of nitrogen relies heavily on various nitrogen transformation reactions performed by diverse and versatile microorganisms [21]. This includes functional genes involved in nitrogen transformations such as nitrogen fixation (*nifH*), nitrification (Archaeal *amoA* (AOA), Bacterial *amoA* (AOB)), and denitrification (*nirK*, *nirS*, *nosZ*). These nitrogen transformations are strongly inhibited in saline environments due to the detrimental effects of salt on functional microorganisms [22]. However, there have been limited studies examining the mechanisms of salt tolerance in quinoa from the perspective of soil microbial communities and nitrogen transformations.

So, to investigate how quinoa enhances microbial resistance to salt stress and influences the expression of nitrogen transformation genes in various saline environments, this study primarily focuses on observing the seedling stage of quinoa (the critical adaptation of quinoa to adverse environmental conditions). Accordingly, this study aims to address the following questions: (1) How does quinoa influence the structure of soil microbial communities under different salinity levels? (2) What are the characteristics of key microbial taxa associated with quinoa in the soil exposed to varying salt stress conditions? (3) What are the effects of quinoa, under salt stress, on microbial nitrogen transformation processes? Overall, this study aims to explore the impact of quinoa on soil microbial communities, keystones, and microbial N transformations under salt stress across three salinity levels. Understanding the succession of soil microbial communities in quinoa is crucial for scientific cultivation and salinization management.

## 2. Material and Methods

### 2.1. Experimental Design and Soil Processing

This study was conducted at the Water-Saving Experimental Station of Hohai University in the Jiangning district (31°91′ N, 118°79′ E), Nanjing, Jiangsu Province, China from October to December 2022. The region is characterized by its subtropical humid monsoon climate. Figure 1 illustrates the average monthly temperature values and fluctuations observed in the greenhouse.

The quinoa seeds utilized in this study were obtained from the Seed Base in Suqian, Jiangsu Province, China. The saline soil used in the experiment was sourced from the demonstration field of Jianfeng Agricultural Industry Co., Ltd., located in the coastal mudflats of Yancheng, Jiangsu Province, China. The original physicochemical properties of the soil are shown in Table 1. A two-factor randomized complete block design with three replications was established, examining six different treatments that varied in the number of cultivation days and salt levels (Figure 2). To alleviate the “pulse effect”, a two-week dark treatment was conducted after accurately introducing sodium chloride (NaCl) to establish controlled salinity levels of 1 g·kg^−1^, 3 g·kg^−1^, and 6 g·kg^−1^, respectively. On 24 October, two quinoa seedlings were planted in each pot. Destructive sampling was conducted at 35 and 70 days of cultivation, respectively. During the cultivation period, the plants were irrigated moderately based on the greenhouse’s humidity and soil conditions. They were nourished with Hogland nutrition solution twice a week, with a dilution ratio of 1/2×.

After the harvest, all samples were divided into two parts: one part was stored at 4 °C for a physicochemical analysis, and the other part was stored at −80 °C for subsequent DNA extraction. The measured soil physicochemical properties include pH, EC (electrical conductivity), the amounts of water-soluble salts in the soil, ammonia–nitrogen (NH_4_^+^_N), and nitrate–nitrogen (NO_3_^−^_N).

### 2.2. DNA Extraction and Quantitative PCR (qPCR)

Soil DNA was extracted using an MP Soil DNA Kit (MP Biomedicals, Santa Ana, CA, USA). The quality of the extracted DNA was assessed using a 2% agarose gel electrophoresis, while the concentration and purity of the DNA were determined using the NanoDrop 2000 spectrophotometer (Thermo Scientific, Waltham, MA, USA). For DNA sequencing, the 16S rRNA (V3–V4) region and ITS sequences were targeted. The primer sequences used for amplicon sequencing can be found in Table 2. After purifying the PCR products, they were sent to Majorbio Bio-Pharm Technology Co., Ltd. (Shanghai, China) for high-throughput sequencing using the Illumina MiSeq PE300 platform of BIOZERON Co., Ltd. (Shanghai, China).

For DNA extraction from each soil sample, the MoBio PowerSoil DNA Isolation Kit (MoBio Laboratories Inc., Carlsbad, CA, USA) was used. A quantitative analysis was conducted to assess the genetic abundance of different nitrogen cycle components, including nitrogen fixation (*nifH*), nitrification (AOA, AOB), and denitrification (*nirK*, *nirS*, *nosZ*). The primer sequences used for the amplicon sequencing of nitrogen transformation genes can be found in Table 3.

### 2.3. Statistical Analysis

The α-diversity and β-diversity analyses were performed using the Majorbio Cloud Platform. The significance of inter-β diversity changes was evaluated using the PERMANOVA test with Bray–Curtis dissimilarity as the differential treatment. STAMP (version 2.1.3) was used to assess the differences in microbial abundance among different operations. The OTU-level co-occurrence network analysis was conducted using R software (version 4.3.1), and the interaction between network nodes was visualized using Gephi software (version 0.10.1). A redundancy analysis (RDA) of soil properties and microbial communities was performed using CANOCO 5 software (Microcomputer Power, Ithaca, NY, USA). In GraphPad Prism 9.5.0.730, the Welch test and Wilcoxon test were used for data with non-normal distributions or no equal variances, as previously described. The related graphics were drawn in R.

## 3. Result and Discussion

### 3.1. Dynamic of Soil Microbial Communities

Changes in microorganisms demonstrate the sensitivity and adaptability of bacteria and fungi in response to variations in the salinity environment. Salt stress resulted in a significant (*p* < 0.05) reduction in the diversity of the soil bacterial communities, as indicated by the Shannon index. Specifically, both the Shannon indexes of CQM1 (7.16) and CQH1 (7.10) were lower than the Shannon index of CQL1 (7.21). However, no significant differences were observed in the Shannon index among the fungal communities in the various treatments, suggesting that bacteria are more susceptible to the adverse effects of salt stress than fungi (Figure 3a). It was discovered that the Shannon index of the bacteria in CQL2, CQM2, and CQH2 increased by 0.58%, 1.33%, and 1.82%, respectively. In CQL2, CQM2, and CQH2, there was a slight decrease in the soil electrical conductivity (EC) and water-soluble salt concentration, but it was not significant (Figure 4). This could be attributed to quinoa’s ability to absorb or accumulate inorganic ions from external sources [35]. Therefore, to adapt to high-salt environments, quinoa establishes a balanced osmotic pressure and indirectly mitigates the detrimental effects of salt on bacterial communities by reducing the salt content in the soil.

The NMDS ordination showed the differences between the bacterial and fungal communities in terms of diversity. Soil salt concentration and cultivation period are two main factors that significantly distinguish bacteria and fungi (Figure 3b). A permutational multivariate analysis of variance (PREMANOVA) further corroborated that the variations in the bacterial communities were mainly driven by soil water-soluble salt conversion (Table 4). However, both the cultivation period and the salinity gradient did not exhibit a significant influence on the alterations in fungal communities. This indicates that fungi are more stable than bacteria under salt stress, and Rath et al. [36] have also observed the same conclusion. It can be explained that variations in bacterial communities under salt stress can affect the penetration of quinoa. Additionally, Otlewska et al. [37] have reported that microbial communities directly/indirectly participate in the osmotic adjustment of plants.

Quinoa benefits from being associated with different microbiological communities, and saline soil is a crucial environmental filter that can promote quinoa to choose soil microbial communities with highly specific traits (Qin et al., 2016 [38]). Figure 3c represents the reduction in bacterial-specific and fungal-specific OUTs in different treatment groups. This means that the functions of quinoa may be altered due to changes in microbial diversity [39]. Numerous studies have highlighted the significance of microbiome diversity in aiding plant adaptation to saline environments [11,40,41]. Toubali and Meddich [11] found that microbial communities promoted the growth of quinoa plants in saline environments and improved soil chemical quality by enhancing enzyme-driven antioxidant and osmotic regulation systems. Different bacterial communities exhibit varying responses to high salt concentrations. Salt stress exerts inhibitory effects on microorganisms such as *Choroflexi*, *Acidobacteriota*, and *Myxococcota*. However, as salt concentrations increase, there is an increasing trend in the abundance of *Bacteroidota*, indicating their ability to thrive in high-salt environments (Figure 5). This discovery aligns with the findings conducted by Rath et al. [42]. These microorganisms exhibiting distinct thresholds in response to salt stress may be selectively favored or recruited by quinoa, leading to the maintenance of a relatively stable population. Similar findings have been observed in studies on rice [43] and *Arabidopsis thaliana* [44]. *Proteobacteria*, *Chloroflexi*, and *Actinobacteria* are the predominant components of soil bacterial communities, and their relative proportions remain relatively constant over time intervals (Figure 6). Additionally, the abundance of Proteobacteria and Actinobacteria is considered a reliable indicator of soil health [45]. In the treatments of CQL1, CQH1, and CQH2, the average relative abundance of *Acidobacteriota*, *Bacteroidota*, and *Gemmatimonadota* was significantly higher, indicating their enrichment in the soil microbial communities (Figure 5). Quinoa and microorganisms engage in mutual interactions that lead to the establishment of specific associations between the plant and certain bacteria at different stages of growth in a salty environment [46]. Importantly, halophytes possess a natural reservoir of salt-resistant microorganisms, which facilitates their growth and development under unfavorable conditions [47]. Moreover, prolonged fluctuations in salinity can contribute to the adaptation of microbial communities [47]. The finding that the average relative abundance of *Planctomycetota* and *Gemmatimonadota* increases with rising salt concentrations and longer growth periods also supports this perspective (Figure 5).

### 3.2. Soil Microbial Co-Occurrence and Keystone Taxa

Co-occurrence networks are frequently used to explore potential connections between microbial populations [22,48,49]. At the genus level, co-occurrence networks were constructed for the bacterial and fungal communities, and keystones were identified to thoroughly investigate the impacts of soil salinity and cultivation days on the soil microbial communities. All genus occurred in the network of the six treatments. Keystones are selected as benchmarks based on the 1% betweenness centrality distribution, considering the top five abundance ratings among these genera [20]. The bacteria keystone taxa in CQL1, CQM1, and CQH1 are seven, nine, and six, respectively, and the bacteria keystone taxa for CQL2, CQM2, and CQH2 are five (Figure 7a). The fungi keystone taxa for CQL1, CQM1, and CQH1 are 10, 12, and 10, respectively, and the fungi Keystone taxa for CQL2, CQM2, and CQH2 are all 10 (Figure 7b). The results demonstrate the consistent response of microbial communities to diverse environmental factors. These genera, through their shared environmental preferences and dispersal capabilities, indicate important classification units that are interconnected and hold significant value [50].

The modularity index of each treatment exceeds 0.4, indicating a characteristic modular structure in the microbial networks of each process (Figure 8). In our study, as the degree of salt stress increased, there were no significant changes observed in the nodes, edges, and network topological indices of both the bacterial and fungal networks (Figure 7). This indicates that salt stress appears to have no negative impact on the interactions among microbial communities. This finding contradicts the conclusion proposed by Xu et al. [51] that salt stress weakens the co-occurrence network of rhizosphere microorganisms. So, we analyzed the sampling abundance of these keystones and found that the bacterial communities are grouped into CQM1 and CQH1 and CQM2 and CQH2, separately from CQL1 and CQL2. Although the layer clustering of fungal communities at the phylum level did not follow a consistent pattern, there was a noticeable fluctuation in sample abundance (Figure 9). This suggests that quinoa may, under salt stress, either promote the emergence of new keystone taxa or alter the abundance of existing keystones to maintain a stable microbial structure [52]. It is also possible that quinoa facilitates the establishment of highly connected and complex networks among soil microorganisms during the early stages to cope with fluctuations in external salt stress conditions [53].

In addition, keystones have been demonstrated to have the proficiency to enhance nutrient conversion and promote plant growth [54]. Keystones, at the phylum level, such as *Proteobacteria* and *Acidobacteria*, are often regarded as copiotrophic microorganisms [19]. The abundant bacterial clusters of *Bacteroidetes* and *Firmicutes* can be regarded as highly salt-resistant species. When the soil is treated with appropriate salinity levels, its relative abundance increases compared to that of untreated soil conditions. Our study yielded similar conclusions to [49] (Figure 9). Changes in community composition at the phylum level may significantly impact the functional characteristics of these communities. Previous studies have found that phylogenetic groups belonging to *Firmicutes* can withstand environmental stress through the formation of their Gram-positive cell walls and spores [55]. Pérez Castro et al. [16] discovered a correlation between *Chloroflexi* and genes related to the metabolism of amino sugars, sugar alcohols, and simple carbohydrates. Therefore, we propose that quinoa enhances its stress resilience by directly influencing microbial abundance and the synthesis of keystones or indirectly participating in microbially mediated nutrient transformations.

### 3.3. Correlations among the Environmental Variables, Bacterial Community, and Nitrogen Functional Genes

A redundancy analysis (RDA) is employed to assess the relationship between the bacterial communities in the different treatment groups and environmental factors, examine the impact of soil properties on community composition at the phylum level, and determine the significance of the association between soil traits and bacterial abundance. The first two axes of the RDA represent 36.96% and 12.92%, respectively, of the total data variation (Figure 10). In bacterial communities, the highest total salinity variation is followed by NH_4_^+^_N (Table 5). In CQL1, *Chloroflexi* predominate, followed by *Acidobacteriota* at CQL2, *Bacteroidota* at CQM1 and CQH1, and *Proteobacteria* at CQM2 and CQH2. In contrast to *Acidobacteriota* and *Proteobacteria*, which are significantly positively correlated with NH_4_^+^_N (*p* < 0.05), *Bacteroidota* and *Chloroflexi* have a positive association with NO_3_^−^_N. Thus, *Bacteroidota* and *Chloroflexi* may have contributed to the conversion and release of NO_3_^−^_N in the quinoa culture for up to 35 days, while *Acidobacteriota* and *Proteobacteria* may have played a more important role in the transformation and discharge of NH_4_^+^_N at 70 days, as confirmed by our measurement of the concentrations of NO_3_^−^_N and NH_4_^+^_N in the soil (Figure 4). Generally, quinoa has the potential to influence the abundance of bacteria during different stages of cultivation, consequently impacting the conversion of nitrogen by different salt sub-processing groups.

The salinity level plays a crucial role in determining the response of nitrogen metabolism to saline stress [1]. We performed the RDA to examine the relationship between the nitrogen transformation genes of the different processing groups and environmental variables (Figure 10b). The results showed that environmental factors accounted for 75.3% of the total variation in nitrogen conversion genes. The interpretation of NH_4_^+^_N and EC changes in the composition of nitrogen-transforming genes were identified as the two primary environmental factors, accounting for 41.6% and 13.8%, respectively, of the variation in the six nitrogen transformer genes (Table 5). The limitations imposed by salinity stress on nitrogen transformation in soil biological systems can be understood by considering the prevalence of nitrogen-transforming genes. On the 35th day of cultivation, the abundance of denitrification genes (*nirS* and *nosZ*) decreased with increasing salt content, while the abundance of nitrification genes (AOA and AOB) showed minimal changes (Figure 11). These factors may be the primary reasons for the increase in NO_3_^−^_N content in the soil (Figure 4). On the 70th day of cultivation, the abundance of denitrification genes (*nirS*, *nosZ*, and *nirK*) showed a positive correlation with salt concentration, while the abundance of nitrification genes (AOA and AOB) exhibited a similar trend (Figure 11). However, there was no significant difference in the NO_3_^−^_N content of the soil, and it exhibited a sharp decline compared to the content at 35 days (Figure 4). We suggest that this could be explained by quinoa actively absorbing Cl^−^ to cope with the prolonged salt stress. Simultaneously, the membrane transport proteins that mediate the transport of Cl^−^ and NO_3_^−^ simultaneously could indirectly facilitate the uptake of NO_3_^−^ from the soil by quinoa [9]. Another possibility is that soil microorganisms are converting NO_3_^−^ into NH_4_^+^ [50]. Due to the relatively low and stable abundance of the nitrogen fixation gene (*nifH*), microorganisms maintain stable ammonium assimilation throughout different cultivation periods (Figure 11). This makes quinoa more inclined to utilize nitrate–nitrogen as a nitrogen source. Miranda-Apodaca et al. [1] have revealed that quinoa employs precise regulation of NO_3_^−^_N and Cl^−^ under salt stress to maintain an optimal root N concentration, enabling effective osmotic regulation. Huang et al. [56] and Zhang et al. [22] also observed similar patterns.

## 4. Conclusions

In conclusion, this study explored the effects of quinoa on soil microbial community structure, changes in key populations, and the regulation of soil nitrogen cycling. The results indicate that soil bacterial communities are more susceptible to salt stress than fungal communities. Quinoa mitigates the adverse effects of salt on bacterial communities by absorbing or accumulating inorganic ions, thereby slightly reducing the soil salinity. Microbial diversity plays an essential role in facilitating quinoa’s adaptation to saline environments. Certain microbial groups, such as *Bacteroidota*, show increased relative abundance under high salt conditions.

It was observed that different environmental factors consistently influence soil microbial community composition, and salt stress does not seem to have a negative impact on microbial interactions. Quinoa may maintain a stable microbial structure by promoting the emergence of new keystones or altering the abundance of existing keystones. Changes at the different phylum levels of these keystones can significantly affect the functional characteristics of microbial communities. Quinoa may enhance its stress tolerance by directly influencing the microbial abundance and the synthesis of keystones or indirectly participating in microbe-mediated nutrient transformations.

There is a correlation between soil bacterial communities and environmental factors in different treatment groups during the period of cultivation. Soil properties significantly influence bacterial community composition at the phylum level. During quinoa cultivation, different salt treatment groups have varying effects on the conversion of NH_4_^+^_N and NO_3_^−^_N. Salt content also influences the abundance of nitrogen transformation genes in the soil. The long-term salt stress adaptation of quinoa may involve the passive absorption of Cl^−^ and the conversion of NO_3_^−^ by soil microorganisms. These results suggest that quinoa adapts and maintains normal growth under salt stress by regulating nitrogen metabolism and ion absorption.

## Figures and Tables

**Figure 1 microorganisms-11-02829-f001:**
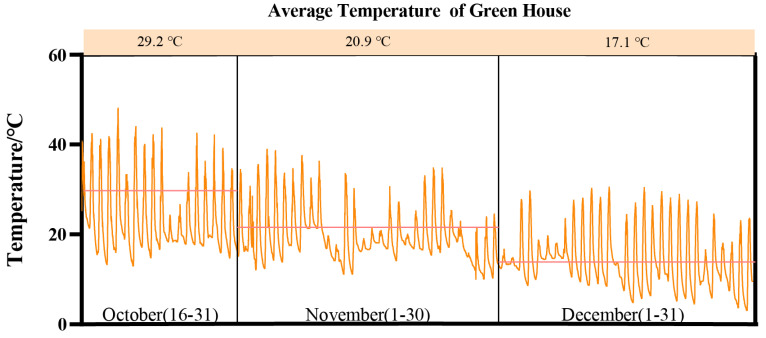
The average temperature of the greenhouse.

**Figure 2 microorganisms-11-02829-f002:**
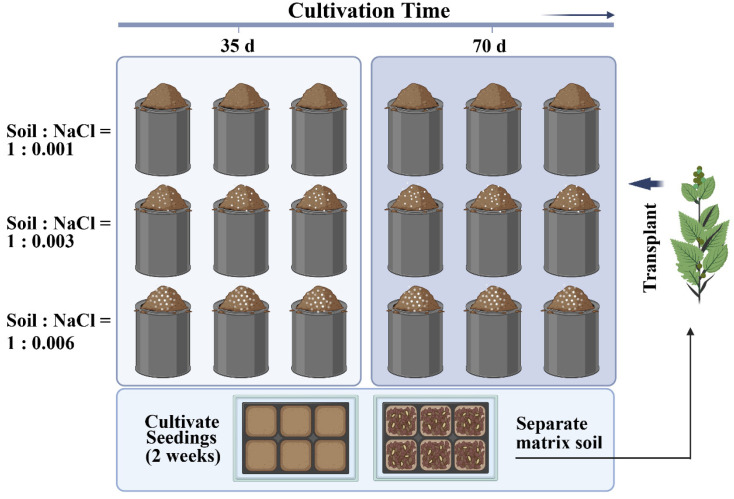
Experiential design. (Untreated and artificially treated soil was filled in each experimental pot; height 22 cm; diameter 18 cm).

**Figure 3 microorganisms-11-02829-f003:**
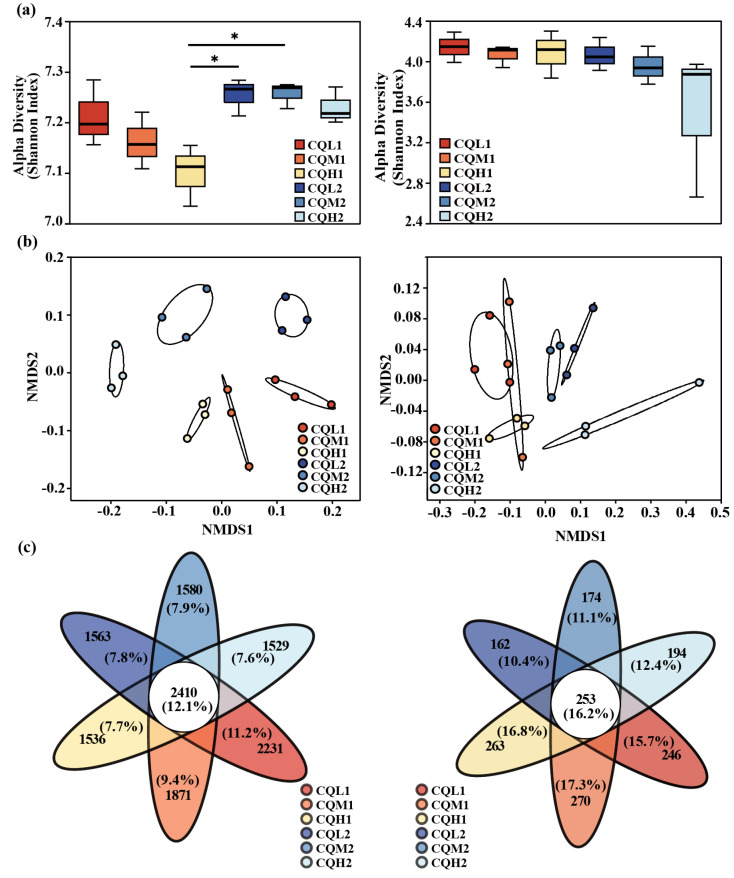
Microbial community diversity and composition of bacteria and fungi. (**a**) Boxplots display the Shannon index, and the difference in α-diversity of bacteria and fungi was detected by the Kruskal–Wallis test; ‘*’ for *p* < 0.05. (**b**) Non-metric multidimensional scaling (NMDS) analysis was based on Bray–Curtis distance for microbial communities, stress < 0.2 (bacteria) and stress < 0.1 (fungi). (**c**) Venn diagrams display the shared and system-specific OTUs of bacteria and fungi.

**Figure 4 microorganisms-11-02829-f004:**
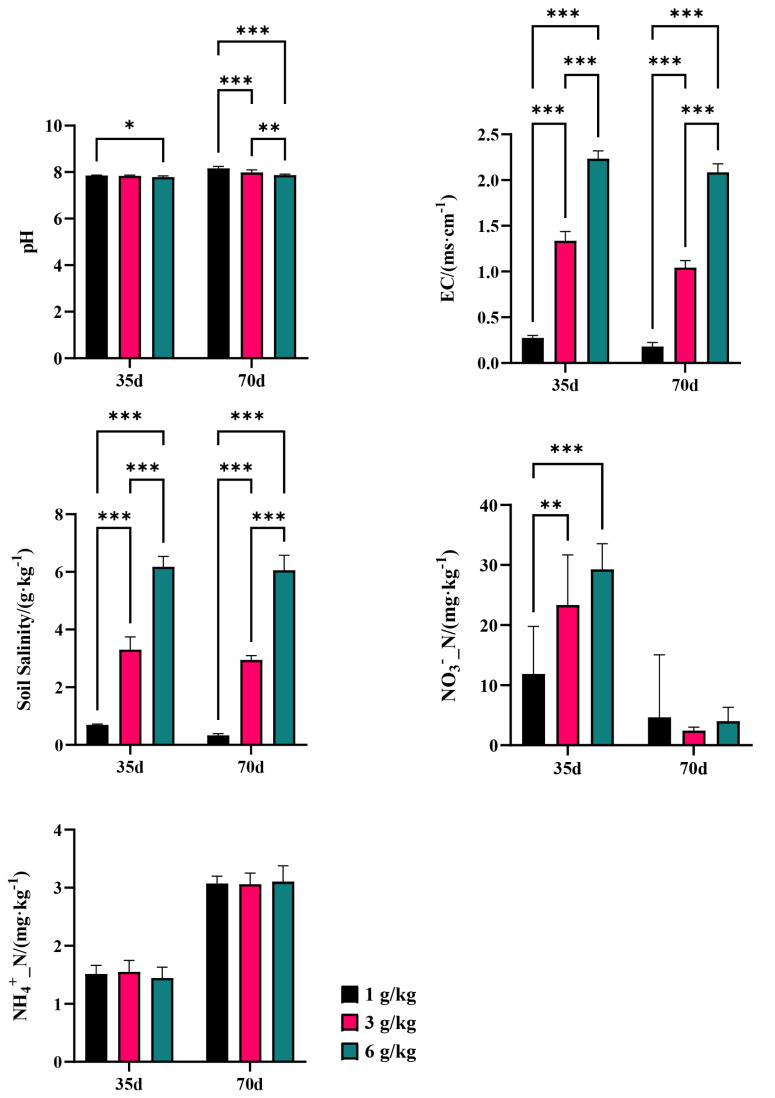
Basic physical and chemical properties of soil; ‘*’ for *p* < 0.05; ‘**’ for *p* < 0.01; ‘***’ for *p* < 0.001.

**Figure 5 microorganisms-11-02829-f005:**
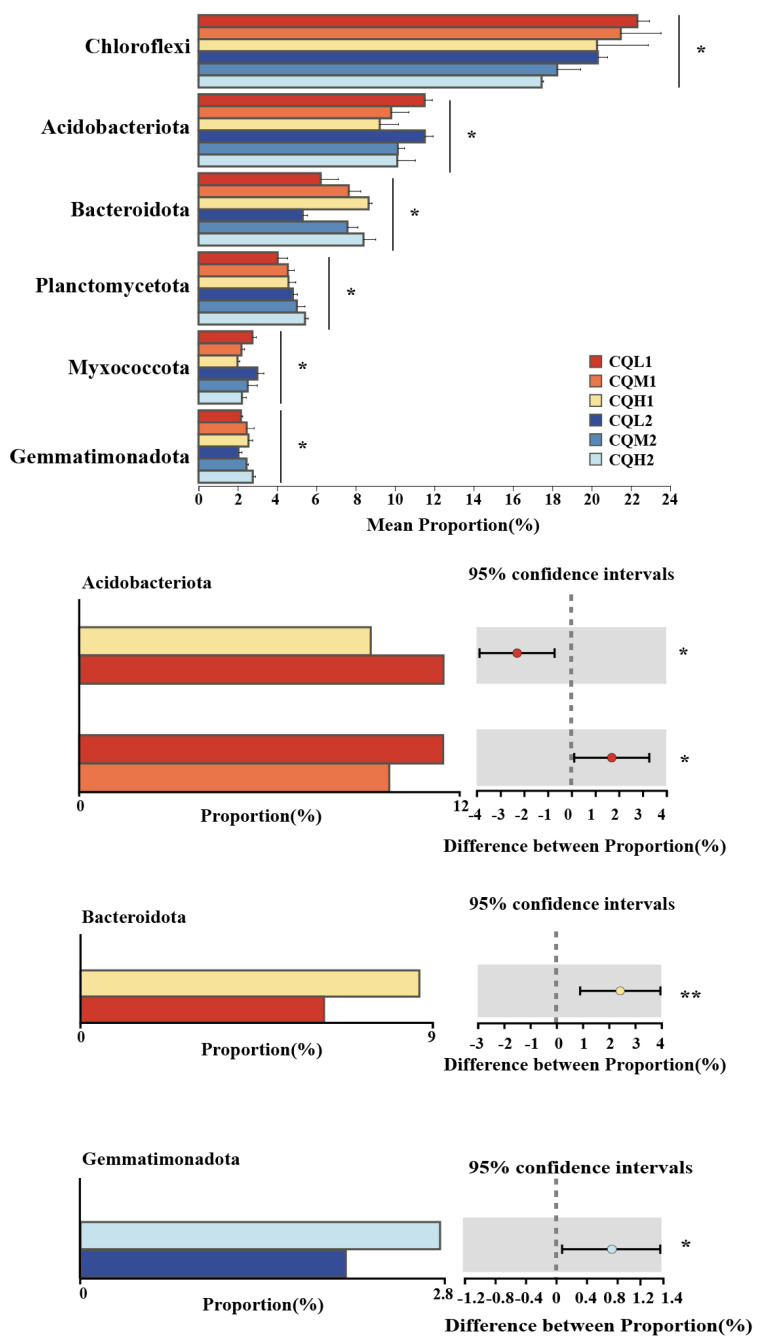
Differentially abundant microbiome at the phylum level; ‘*’ for *p* < 0.05; ‘**’ for *p* < 0.01.

**Figure 6 microorganisms-11-02829-f006:**
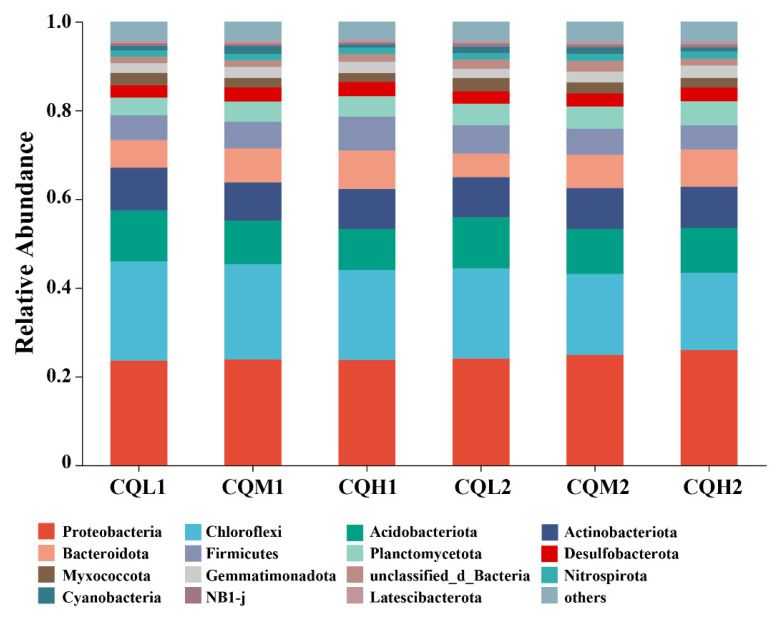
Relative abundance of microbial communities at the phylum level in six treatments.

**Figure 7 microorganisms-11-02829-f007:**
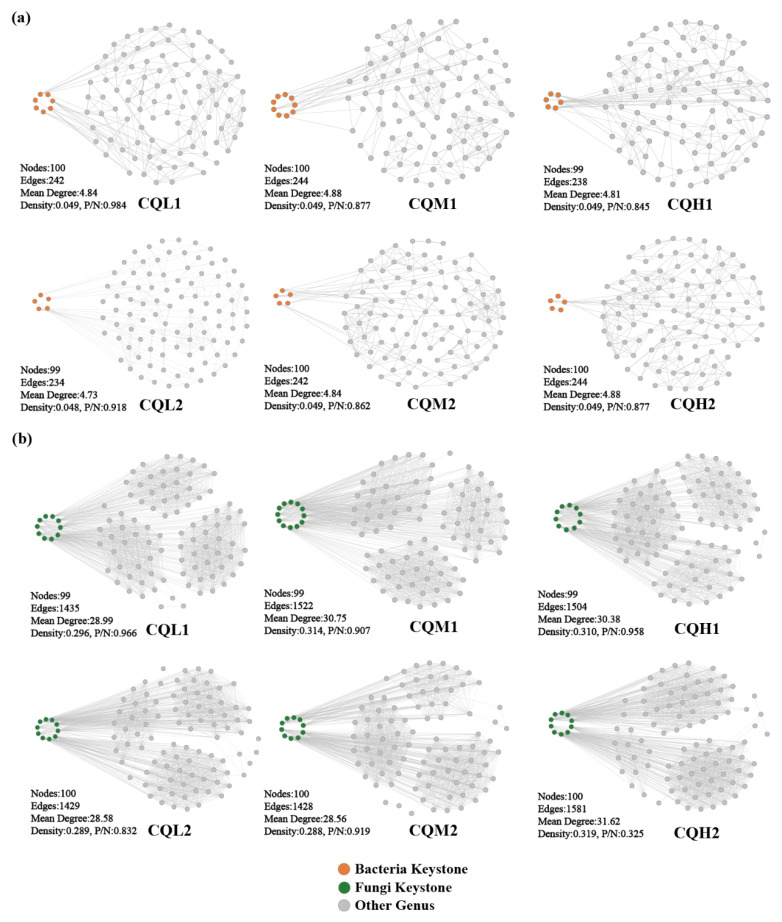
Co-occurrence networks of microbial communities in different treatment groups at the genus level. Topological properties of co-occurrence networks were indicated under each network, including the number of nodes, number of edges, mean degree, density, and the ratio of positive to negative interactions (P/N) of the whole network. (**a**) Networks of bacterial communities, (**b**) Networks of fungal communities.

**Figure 8 microorganisms-11-02829-f008:**
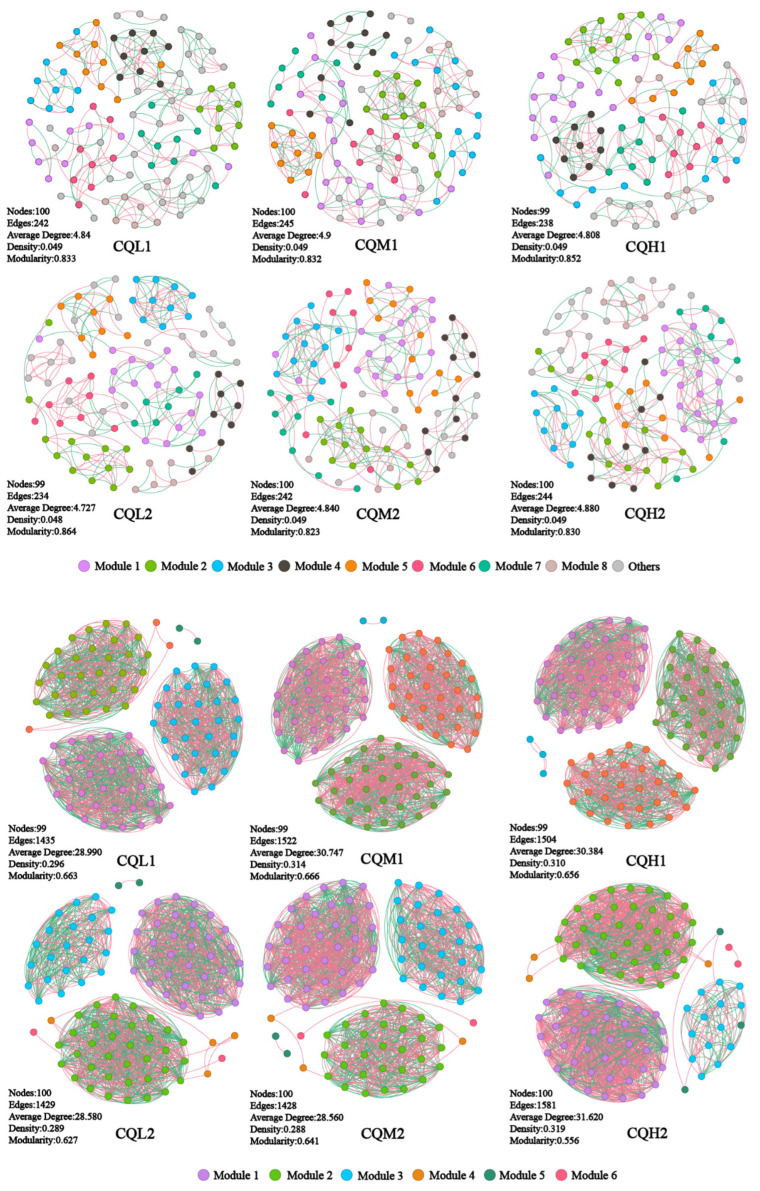
Co-occurrence networks of the microbial communities in the different treatment groups.

**Figure 9 microorganisms-11-02829-f009:**
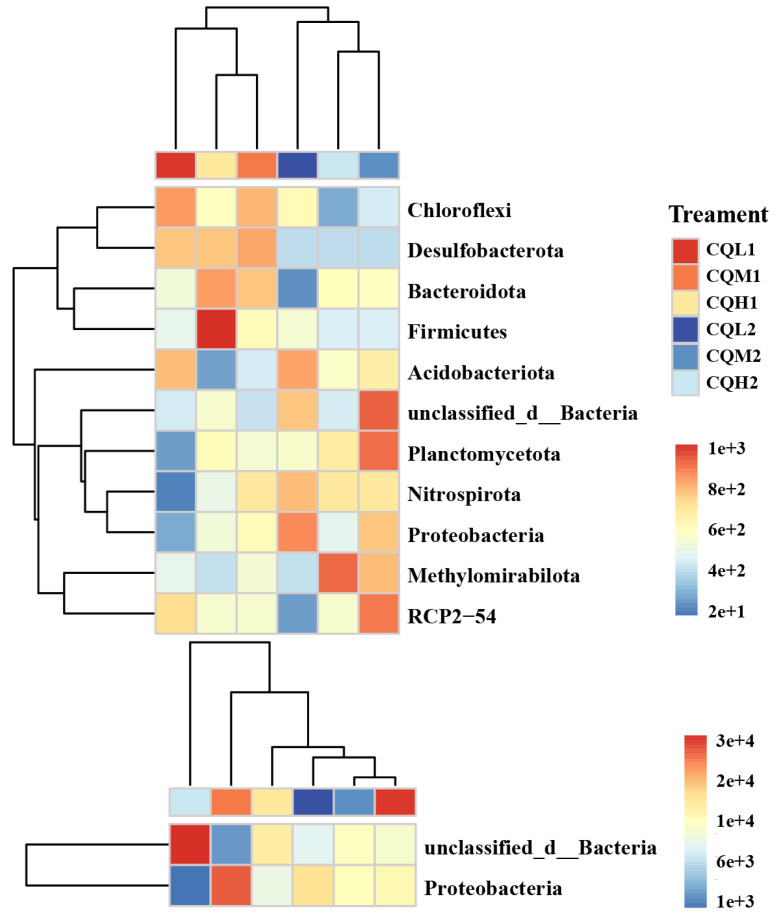
Significant responses of keystone taxa. These keystone taxa are further annotated at the phylum level in the heatmap. Calculating the distance between samples using Euclidean Distance, average linkage clustering determines the relationship between the samples.

**Figure 10 microorganisms-11-02829-f010:**
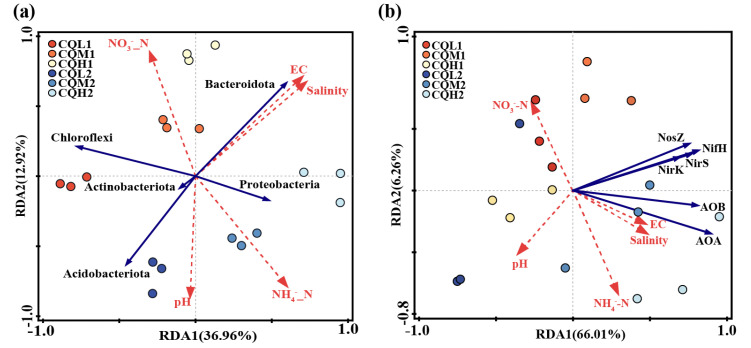
Redundancy analysis based on the relationships between environmental variables, top 5 phyla (**a**), and nitrogen functional genes (**b**) during the overall experimental process. The red arrow represents environmental factors, while the blue arrows respectively represent phyla and nitrogen functional genes.

**Figure 11 microorganisms-11-02829-f011:**
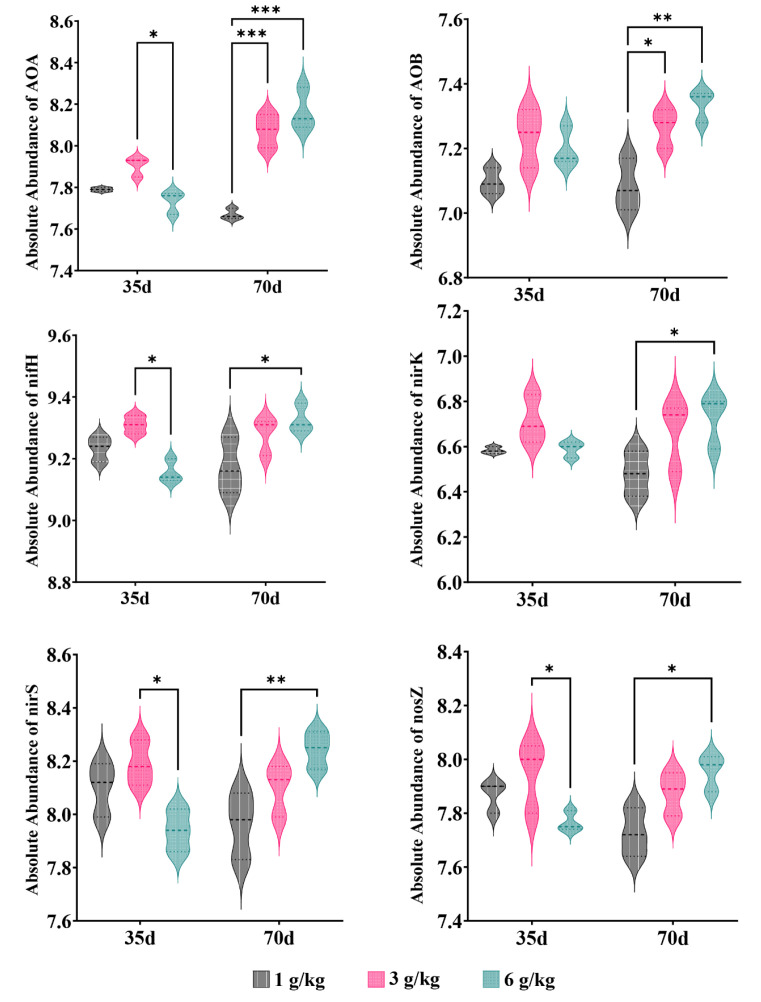
Relative abundance of nitrogen genes; ‘*’ for *p* < 0.05; ‘**’ for *p* < 0.01; ‘***’ for *p* < 0.001.

**Table 1 microorganisms-11-02829-t001:** The initial rationalization properties of the tested soil samples.

pH	EC (ms·cm^−1^)	Salinity (g·kg^−1^)	TOC (%)	TN (g·kg^−1^)	AP (mg·g^−1^)	SWC (%)
7.35 ± 0.03	0.28 ± 0.01	0.72 ± 0.10	1.47 ± 0.03	0.94 ± 0.03	23.89 ± 0.20	27.87

**Table 2 microorganisms-11-02829-t002:** Primer of amplicon sequencing.

Amplicon Sequencing	Primer IDs	Forward Primer Sequence (5′–3′)	Reference
16S rRNA	515F	GTGNCAGCMGCCGCGGTAA	Quince et al. (2011) [23]
	907R	CCGYCAATTYMTTTRAGTTT	Lane et al. (1991) [24]
ITS	ITS1F	CTTGGTCATTTAGAGGAAGTAA	Gurr et al. (1991) [25]
	ITS2R	GCTGCGTTCTTCATCGATGC	Edgar et al. (2013) [26]

**Table 3 microorganisms-11-02829-t003:** Genes quantified using qPCR.

Gene	Primer IDs	Primer Sequence (5′–3′)	Reference
Archaeal *amoA*	Arch-amoAF	STAATGGTCTGGCTTAGACG	Tourna et al. (2008) [27]
	Arch-amoAR	GCGGCCATCCATCTGTATGT	Francis et al. (2005) [28]
Bacterial *amoA*	amoA-1F	GGGGTTTCTACTGGTGGT	Rotthauwe et al. (1997) [29]
	amoA-2R	CCCCTCKGSAAAGCCTTCTTC	
*nifH*	nifH-F	AAAGGYGGWATCGGYAARTCCACCAC	Rösch and Bothe (2005) [30]
	nifH-Rb	TTGTTSGCSGCRTACATSGCCATCAT	
*nirS*	cd3AF	GTSAACGTSAAGGASACSGG	Michotey et al. (2000) [31]
	R3cd	GASTTCGGRTGSGTCTTGA	Kandeler et al. (2006) [32]
*nirK*	nirK 1F	GGMATGGTKCCSTGGCA	Braker et al. (1998) [33]
	nirK 5R	GCCTCGATCAGRTTRTGGTT	
*nosZ*	nosZ2F	CGCRACGGCAASAAGGTSMSSGT	Henry et al. (2006) [34]
	nosZ2R	CAKRTGCAKSGCRTGGCAGAA	

**Table 4 microorganisms-11-02829-t004:** The effects of soil salinity and incubation time on the differentiation of soil microbial communities based on permutational multivariate analysis of variance (PERMANOVA).

Microbial Community	Phylum	Salinity	Day
Bacteria	R^2^	0.236	0.129
	P	0.002	0.031
	Sig.	**	*
Fungi	R^2^	0.195	0.086
	P	0.085	0.243
	Sig.	-	-

Note: ‘*’ for *p* < 0.05, ‘**’ for *p* < 0.01.

**Table 5 microorganisms-11-02829-t005:** Significant physicochemical parameters and explanations of RDA of (a) top 5 phyla and (b) nitrogen functional genes.

Variable	Explains (%)	F	P
Salinity	24.8	5.3	0.006 ^**^
NH4^+^_N	22.4	6.4	0.014 ^*^
(a)			
NH4^+^_N	41.6	12.8	0.002 ^**^
EC	13.8	6.6	0.018 ^*^
(b)			

Note: ‘*’ for *p* < 0.05, ‘**’ for *p* < 0.01.

## Data Availability

Data are contained within the article.

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
