# Peer review of "The Salinity Survival Strategy of Chenopodium quinoa: Investigating Microbial Community Shifts and Nitrogen Cycling in Saline Soils"

_microorganisms, 2023, doi:10.3390/microorganisms11122829_

Round 1

Reviewer 1 Report

Comments and Suggestions for Authors

More researches should be following this paper work because its very important point reflecting the correlation between soil bacterial communities and environmental factors 346 in different treatment groups during the period of cultivation. Soil properties significantly 347 influence bacterial community composition at the phylum level. During quinoa cultiva- 348 tion, different salt treatment groups have varying effects on the conversion of NH4 +_N and 349 NO3 -_N. Salt content also influences the abundance of nitrogen transformation genes in 350 the soil. Long-term salt stress adaptation of quinoa may involve passive absorption of Cl- 351 and the conversion of NO3 - by soil microorganisms. These results suggest that quinoa 352 adapts and maintains normal growth under salt stress by regulating nitrogen metabolism 353 and ion absorption.

Certainly, this research was well prepared in sound scientific language, with the exception of some linguistic corrections. The materials and methods used in the work were clearly presented, the results were presented clearly, and the results were well discussed. The figures and tables were expressive of what was included in the research. The importance of the research is due to the possibility of producing Quinoa in saline soils, and this is one of the new and promising areas in the field of nutrition and health security.  

This research proved that Quinoa has been widely cultivated for its nutritional value and exceptional ability to tolerate high levels of salt, which represents a promising solution to the agricultural challenges resulting from salt stress. The mechanisms of salt tolerance of quinoa were studied, and changes in microbial community structure and abundance of nitrogen conversion genes were assessed under three salinity levels (1 g kg-1, 3 g kg-1, and 6 g kg-1) at two distinct time points (35). and 70 days). 

The results were promising as they demonstrated the positive effect of quinoa on soil microbial community structure, changes in key populations, and the regulatory role in soil nitrogen cycling under salt stress. Choroflexi, Acidobacteriota, and Myxococcota were inhibited by increasing salinity, while the relative abundance of Bacteroidota increased. Proteobacteria and Actinobacteria show relatively stable abundances across time and salinity levels. 

The researchers demonstrated that Quinoa can synthesize or change the composition of essential species or promote the establishment of highly complex microbial networks to cope with fluctuations in external salt stress environments. Furthermore, quinoa maintained the nitrogen (N) cycle by downregulating dehydration genes.  

This study paves the way for future research on Quinoa regulation, enhancing soil microbial communities and N conversion in saline environments. 

Comments on the Quality of English Language

Minor English editing

Author Response

Dear Reviewer,

We appreciate your suggestions. We have made revisions to address potential 
ambiguities in the text and replaced certain words and expressions with more 
appropriate ones. These changes will not influence the content and framework of the paper. We have also conducted a thorough grammar check throughout the entire document. The modified sections of the text have been clearly highlighted in yellow.

Reviewer 2 Report

Comments and Suggestions for Authors

The study examines the microbial Community shifts and nitrogen cycling in saline soils of Quinoa. Overall, the study is intriguing and fits well within the scope of the journal, Microorganisms, and will appeal to its readership. However, the novelty and/or contribution of this study to its field needs to be highlighted in the Introduction section.

Author Response

Dear Reviewer,

Thanks for your comments. We have added the innovative aspects of this paper to the final paragraph of the introduction, with blue text.

Reviewer 3 Report

Comments and Suggestions for Authors

The paper „The Salinity Survival Strategy of Chenopodium Quinoa: Investigating Microbial Community Shifts and Nitrogen Cycling in Saline Soils” is current and very well structurate.

The authors present a study primarily focused on observing the seedling stage of quinoa (critical adaptation of quinoa to adverse environmental conditions). The study of the authors was focused to explore the effects of quinoa on soil microbial community, keystones, and microbial N- transformation under salt stress in three salinity levels.

The experimental results obtained by the authors, indicate that soil bacterial communities are more susceptible to salt stress than fungal communities. Also the authors show that quinoa mitigates the adverse effects of salt on bacterial communities by absorbing or accumulating inorganic ions, thereby slightly reducing soil salinity.

Also the authors prove that soil properties significantly influence bacterial community composition at the phylum level.

I propose to publish the paper in present form.

Author Response

Dear Reviewer,

We are thankful to you for your recognition of our research work.